# A Ferritin Nanoparticle-Based Zika Virus Vaccine Candidate Induces Robust Humoral and Cellular Immune Responses and Protects Mice from Lethal Virus Challenge

**DOI:** 10.3390/vaccines11040821

**Published:** 2023-04-10

**Authors:** Aryamav Pattnaik, Bikash R. Sahoo, Lucas R. Struble, Gloria E. O. Borgstahl, You Zhou, Rodrigo Franco, Raul G. Barletta, Fernando A. Osorio, Thomas M. Petro, Asit K. Pattnaik

**Affiliations:** 1School of Veterinary Medicine and Biomedical Sciences, University of Nebraska-Lincoln, Lincoln, NE 68583, USA; apattna@purdue.edu (A.P.); bsahoo2@unl.edu (B.R.S.); yzhou2@unl.edu (Y.Z.); rodrigo.franco@unl.edu (R.F.); rbarletta@unl.edu (R.G.B.); fosorio1@unl.edu (F.A.O.); 2Nebraska Center for Virology, University of Nebraska-Lincoln, Lincoln, NE 68583, USA; tpetro@unmc.edu; 3The Eppley Institute for Cancer and Allied Diseases, Fred & Pamela Buffet Cancer Center, University of Nebraska Medical Center, Omaha, NE 68198, USA; lucas.struble@unmc.edu (L.R.S.); gborgstahl@unmc.edu (G.E.O.B.); 4Center for Biotechnology, University of Nebraska-Lincoln, Lincoln, NE 68583, USA; 5Redox Biology Center, University of Nebraska-Lincoln, Lincoln, NE 68583, USA; 6Department of Oral Biology, University of Nebraska Medical Center, Lincoln, NE 68583, USA

**Keywords:** Zika virus, envelope protein domain III, ferritin, nanoparticles, neutralizing antibody response, cell-mediated response

## Abstract

The severe consequences of the Zika virus (ZIKV) infections resulting in congenital Zika syndrome in infants and the autoimmune Guillain–Barre syndrome in adults warrant the development of safe and efficacious vaccines and therapeutics. Currently, there are no approved treatment options for ZIKV infection. Herein, we describe the development of a bacterial ferritin-based nanoparticle vaccine candidate for ZIKV. The viral envelope (E) protein domain III (DIII) was fused in-frame at the amino-terminus of ferritin. The resulting nanoparticle displaying the DIII was examined for its ability to induce immune responses and protect vaccinated animals upon lethal virus challenge. Our results show that immunization of mice with a single dose of the nanoparticle vaccine candidate (zDIII-F) resulted in the robust induction of neutralizing antibody responses that protected the animals from the lethal ZIKV challenge. The antibodies neutralized infectivity of other ZIKV lineages indicating that the zDIII-F can confer heterologous protection. The vaccine candidate also induced a significantly higher frequency of interferon (IFN)-γ positive CD4 T cells and CD8 T cells suggesting that both humoral and cell-mediated immune responses were induced by the vaccine candidate. Although our studies showed that a soluble DIII vaccine candidate could also induce humoral and cell-mediated immunity and protect from lethal ZIKV challenge, the immune responses and protection conferred by the nanoparticle vaccine candidate were superior. Further, passive transfer of neutralizing antibodies from the vaccinated animals to naïve animals protected against lethal ZIKV challenge. Since previous studies have shown that antibodies directed at the DIII region of the E protein do not to induce antibody-dependent enhancement (ADE) of ZIKV or other related flavivirus infections, our studies support the use of the zDIII-F nanoparticle vaccine candidate for safe and enhanced immunological responses against ZIKV.

## 1. Introduction

The re-emergence of the Zika virus (ZIKV) in the Americas, and its association with microcephaly as well as other neurological symptoms in infants and Guillain–Barre syndrome in adults, poses a significant threat to human health [1,2,3]. While the incidence of ZIKV infections has declined substantially in recent years, the potential for its re-emergence or re-introduction, particularly in mosquito-infested regions, remains high. Therefore, the development of clinically approved antivirals and vaccines with high efficacy is a top priority.

ZIKV, a member of the *Flaviviridae* family, has a positive-sense RNA genome of ~10.8 kb that encodes a single open reading frame (ORF). In infected cells, the genome is first translated into a polyprotein that is processed by cellular and viral proteases to yield three structural proteins, capsid (C), pre-membrane (prM) and envelope (E), and seven non-structural (NS) proteins: NS1, NS2A, NS2B, NS3, NS4A, NS4B and NS5 [4,5]. While the NS proteins are involved in the replication of the genome, assembly of progeny virions, manipulation of cellular signaling pathways and evasion of host antiviral responses, the structural proteins are mainly responsible for virus attachment and entry into host cells as well as virus particle formation [4,6]. Similar to other flavivirus E proteins, the ZIKV E protein exists as a homodimer and is composed of three domains (DI, DII and DIII) [7]. The full-length E protein as well as the DIII region of flaviviruses including ZIKV are major targets for the development of neutralizing antibodies [8,9,10,11].

Several vaccine candidates are in clinical testing phases [12,13,14]; however, none has been approved for use in preventing ZIKV infections. While live attenuated vaccine candidates are more efficacious compared to those based on inactivated viruses, or recombinant subunits, they have the potential to revert to virulent phenotypes causing illnesses [15,16]. The inactivated viruses, or recombinant subunit-based vaccines, on the other hand, are relatively safe except when used in some immunocompromised individuals [15,16]. In addition, inactivated virus-based vaccines or subunit vaccines can be less efficient in promoting protective immune responses [17]. Vaccine platforms involving self-assembling protein nanoparticles with the ability to display multiple copies of an antigen on the surface have emerged as alternatives platforms. Such platforms can be instrumental in overcoming the issues associated with the above-mentioned vaccine platforms and aid in the development of safer and more efficacious vaccines [15,16]. Furthermore, the observations that our immune system responds more efficiently to immunogens with sizes in the nanometer range [18,19] suggest that the efficacy of nanoparticles displaying multiple copies of an antigen as vaccines is likely to exceed that of a soluble monomeric antigen. Since repetitive array of a protein antigen improves immunogenicity and leads to increased antibody responses [20,21], nanoparticles have become desirable platforms for generating efficacious vaccines. The advantages of nanoparticle-based vaccines as opposed to soluble monomeric vaccines lie in the presentation of multiple copies of an immunogen on the surface of the nanoparticle. This platform can deliver an ordered array of the antigen that can result in stronger interactions with multiple B-cell receptors (BCRs); this is critical for downstream signaling for potent B cell activation as well as antibody maturation [16,22,23]. Furthermore, nanoparticles smaller than 100 nm are readily taken up by dendritic cells [24] that then migrate to lymph nodes for presentation of the antigen to trigger T cell immune responses [25]. Thus, a nanoparticle vaccine candidate displaying a viral antigenic protein can result in more efficient interaction with B-cell receptors, a crucial step in B cell-induced immune responses that are multitudes of magnitude higher than traditional vaccines. The suitability of nanoparticle-based vaccines for large scale production also adds to the growing interest in this approach.

Several naturally occurring proteins can self-assemble to form stable and structurally organized nanoparticles with optimum dimensions that make them suitable to serve as vaccine delivery platforms [16,26,27]. One such example is ferritin, a protein found in all living organisms that store iron, thereby protecting cells from reactive oxygen species generated by exposure to excess iron [16,28]. The mammalian and insect ferritin nanoparticles are composed of 24 subunits of any combination of heavy and light chains and are secreted from cells [29]. However, the bacterial ferritin nanoparticle is composed of 24 identical subunits that are organized to form an octahedral structure and reported to be resistant to thermal and chemical degradation [28,30]. The first report of the use of ferritin nanoparticles as a vaccine platform demonstrated that influenza virus hemagglutinin (HA) could be displayed on the surface of the nanoparticle in its native trimeric form resulting in significantly higher induction of neutralizing antibodies, thus providing evidence for more potent immune responses [31]. This strategy has been used successfully in the development of a universal influenza vaccine, which has shown promising results with heterotypic protection [32]. The ferritin nanoparticle approach has also been used to develop vaccines against several viral pathogens, such as the Epstein–Barr virus [33], human immunodeficiency virus (HIV) [34,35,36], hepatitis C virus (HCV) [37,38], human respiratory syncytial virus [39] and ZIKV [40]. Most recently, ferritin nanoparticle-based vaccine candidates have been developed against SARS-CoV-2 using the full-length spike (S) protein or its receptor binding domain (RBD) [41,42,43,44,45,46,47], which show robust and persistent antibody responses with long-term memory.

In this study, we sought to generate a vaccine candidate with only the domain III (DIII) of ZIKV E protein as the immunogen in association with ferritin. DIII was selected to avoid regions of the viral E protein that are known to induce non-neutralizing antibodies that can promote antibody-dependent enhancement (ADE) of infection of either ZIKV or other flaviviruses such as dengue virus (DENV) [48,49,50,51,52]. Transfection of a plasmid constructs encoding ferritin fused in-frame at its amino-terminus with DIII of ZIKV E protein in mammalian cells, resulting in production of the recombinant fusion protein that was secreted from the cells. The purified fusion protein, zDIII-F, under transmission electron microscopy was seen to form typical nanoparticle-like structures. The zDIII-F recombinant protein (hereafter called the nanoparticle) was stable and retained its organized structure for prolonged periods of time. In animal studies, a single immunization with this nanoparticle vaccine candidate induced robust neutralizing antibody response compared to the soluble DIII antigen alone and conferred protection against lethal ZIKV challenge. Significantly higher levels of cellular immune responses with increased frequency of CD4^+^/IFN-γ^+^ as well as CD8^+^/IFN-γ^+^ T cells were seen in immunized mice. Furthermore, sera obtained from immunized mice protected naïve animals from lethal challenge with ZIKV. The results provide support for the use of the ferritin nanoparticle, displaying DIII as a vaccine candidate for ZIKV for further development and clinical evaluation.

## 2. Materials and Methods

### 2.1. Ethics Statement

This study was performed as per the recommendations in the Guide for the Care and Use of Laboratory Animals of the National Institutes of Health. The protocol (# 1323) was approved by the Institutional Animal Care and Use Committee (IACUC) of the University of Nebraska–Lincoln (UNL). Animals were housed in the Life Sciences Annex building at the University. All procedures were conducted under anesthesia with isoflurane, and all efforts were made to minimize animal suffering.

### 2.2. Cells and Viruses

Vero, the African green monkey kidney cells (Cercopithecus aethiops, CCL-81), obtained from ATCC (Manassas, VA, USA), were grown and maintained in Dulbecco’s modified Eagle medium (DMEM) (Invitrogen, Carlsbad, CA, USA) supplemented with 10% fetal bovine serum (heat-inactivated) (FBS; HyClone Laboratories, Logan, UT, USA) and 1% penicillin/streptomycin (PS) (Invitrogen, Carlsbad, CA, USA) in a humidified chamber with 5% CO_2_ at 37 °C. Monolayers of HEK293T cells were also grown similarly in DMEM medium. HEK293-Freestyle (HEK293-F) cells (Cat# R79007) obtained from ThermoFisher Scientific were grown in FreeStyle™ 293 Expression Medium (Cat# 12338018) in sterile conical flasks on a shaking (at 140 rpm) platform in a humidified incubator with 8% CO_2_ at 37 °C.

The infectious-clone-derived recombinant MR766 (rMR) virus has been described previously [53]. Other ZIKV isolates (PRVABC59 and Mex1-7) have been reported previously [53]. Virus stocks were prepared by infecting confluent monolayers of Vero cells at a multiplicity of infection (MOI) of 0.1 plaque-forming unit (PFU) per cell and incubating in virus growth medium (VGM) [DMEM containing 2% FBS, 20 mM hydroxyethyl piperazine ethane sulfonic acid (HEPES), 1 mM sodium pyruvate, non-essential amino acids and 1% PS)] for four days, as described previously [53,54,55].

### 2.3. Reagents and Antibodies

Restriction enzymes, DNA-modifying enzymes, the Q5 High Fidelity PCR kit and the ProtoScript II First Strand cDNA synthesis kit were obtained from New England Biolabs (Ipswich, MA, USA). SuperScript II was obtained from Invitrogen (Carlsbad, CA, USA).

Anti-flavivirus monoclonal antibody D1-4G2-4-15, which reacts with ZIKV E protein, was obtained from EMD Millipore (Billerica, MA, USA). Secondary antibodies were obtained from Sigma (St. Louis, MO, USA) and Invitrogen. Oligonucleotide primers and probes for DNA amplification and quantitative PCR (qPCR) were obtained from Sigma and IDT (Coralville, IA, USA). Anti-ZIKV E antibody was obtained from Genetex (Cat # GTX1333325). Anti-ZIKV E DIII monoclonal antibody (ZV-57) was a kind gift from Michael Diamond, Washington University School of Medicine, St. Louis, MO, USA.

### 2.4. Plasmid Constructs, Protein Expression, Western Blotting and Purification of zDIII-F Nanoparticles

pIRES2-eGFP vector (CloneTech Laboratories, now part of TaKaRa Bio, San Jose, CA USA) was used for expression of the recombinant protein. A DNA fragment corresponding to *Helicobacter pylori* ferritin sequences (encoding amino acid residues 5 to 167) along with the MR766 ZIKV domain III residues 303–404 (zDIII-F) and a secrecon secretion signal sequence [56] were synthesized using GeneScript (Piscataway, NJ, USA). The expression cassette was codon optimized prior to gene synthesis for efficient expression in mammalian cells. The DNA fragment was cloned using the NheI and NotI sites in the pIRES2-eGFP vector under the control of a CMV promoter and the authenticity of the clone was verified by using restriction enzyme digestion and nucleotide sequencing. The DIII coding region along with that of the secretion signal sequence were separately PCR-amplified and cloned using the same restriction sites as described above in the pIRES2-eGFP vector for expression of the protein for use as soluble DIII protein.

HEK293T cells were initially transfected with the plasmid DNA using Lipofectamine2000 (InVitrogen). Cell culture supernatants and cell lysates were examined for expression and secretion of the protein. Once confirmed by Western blotting, scaled-up production of zDIII-F was initiated using HEK293-F suspension culture cells. These cells were seeded at a density of 5–7 × 10^5^ cells per ml one day prior to transfection. The cells were checked for viability using trypan blue dye exclusion assay. DNA transfection was performed if the cells were >90% viable and the cell density was ~1 × 10^6^ cells per ml. For transfection, plasmid DNA at 1 μg per 1 million cells was mixed with the appropriate volume of OptiMEM containing 293Fectin (ThermoFisher Cat# 12347019) or polyethyleneimine (PEI). The DNA and 293Fectin (1:4) or PEI mixture were incubated at RT for 20 min to facilitate the formation of DNA lipid complexes and was added to the cells. The cells were incubated for 4 days on a shaking platform following which they were pelleted at 8000× *g* for 20 min at 4 °C. Cell pellets and the clarified culture media were checked for protein expression by using SDS-PAGE and Western blotting. Culture supernatants were used for purification of zDIII-F nanoparticles by ion-exchange and size exclusion chromatography.

Western blotting was performed as described previously [55]. The primary antibody (ZV-57) was used at a dilution of 1:1000 in 5% non-fat milk in TBS-T and the membranes were incubated at 4 °C overnight. The membrane was washed three times in TBS-T and incubated for 2 h at room temperature with HRP-conjugated secondary goat antimouse (Sigma Cat# 12-349) antibody at a dilution of 1:10,000, in 5% non-fat milk in TBS-T. Blots were washed three times in TBS-T, 5 min each, and then ECL was added to the membrane and the blot was developed using the Bio-Rad imager.

Following confirmation of expression and secretion of the protein, the clarified culture medium was filtered through a 0.2 μm filter and buffer exchanged with TN50 buffer containing 20 mM Tris pH 7.5 and 50 mM NaCl. The buffer exchange was performed using a tangential flow filtration system with a Pellicon^®^ 2 Mini Cassette with an Ultracel^®^ 30 kDa Membrane (Millipore). Protein was then purified from this buffer exchanged media using a 5 mL HiTrap Q FF column (GE Healthcare) and an ÄKTApure system (GE Healthcare). The column was rinsed with 10 column volume (CV) of TN50 buffer, and then a 20 CV elution gradient from 50 to 1000 mM NaCl was utilized for elution of the protein. Peak fractions were tested for the presence of the protein using SDS-PAGE and Coomassie blue staining, and fractions containing the ZDIII-F protein were pooled and concentrated using an Amicon Ultracentrifugal filter unit with a 100 kDa cut off (Millipore). This was followed by separation of the pooled fractions using size exclusion chromatography (SEC) using a Superdex 200 16/600 pg column (GE healthcare). The elution of the purified protein was performed in PBS and fractions were examined by using SDS-PAGE and Coomassie staining. The eluted protein fractions were stored at −80 °C until further use. For thermostability analysis, aliquots of the purified zDIII-F protein were stored at 4 °C for various lengths of time and examined by using SDS-PAGE and Coomassie staining.

### 2.5. Transmission Electron Microscopy

A 30 µL droplet of specimen was placed on parafilm and the surface of a carbon-formvar-coated copper grid was placed upside down to contact the samples for 1–2 min. After excess specimen was wicked off the grid by touching a piece of filter paper to the edge of the grid surface, the grid with the sample side up was air-dried for ~2 min. The grid was placed upside down on to a drop (30 µL) of 1% phosphotungstic acid solution for ~2 min. Excess stain solution was wicked off the grid following the staining and air-dried for at least 30 min. Samples were examined and different magnifications of images were collected using a Hitachi H7500 TEM (at 80 KV). 

### 2.6. zDIII-F Vaccination and Challenge Studies in Mice

Three-week-old interferon (IFN) α/β receptor knockout mice (A129) were purchased from the Jackson Laboratory (Bar Harbor, ME, USA) and acclimatized for 4 days in the animal facility at the University of Nebraska–Lincoln. Purified zDIII-F nanoparticles (10 µg) or the soluble ZIKV DIII protein in molar equivalents (4 µg) in 100 µL of PBS mixed with 100 µL of Addavax (Invivogen Cat# vac-adx-10) were administered into mice (n = 6 or 7) using a 22-gauge needle by the subcutaneous (s.c.) route. PBS was administered to mice in the control group. Following this, the mice were observed daily for 28 days post-vaccination and then challenged with a lethal dose of ZIKV (rMR, 10,000 pfu/mouse) by the s.c. route as described previously [57]. The mice were kept under observation for the development of disease symptoms for 7–10 days. Blood samples were collected by retro-orbital puncture under anesthesia on days 2, 4 and 6 post-challenge to determine viral genome copies. For the passive transfer experiment, four-week-old A129 mice (n = 10) were injected s.c. with 200 μL of naïve (PBS-treated) or zDIII-F immunized mice serum. Four hours after serum transfer, the mice were given a lethal dose of ZIKV (rMR, 10,000 pfu/mouse) s.c. and were monitored for the development of disease symptoms for 7–10 days. Blood samples were collected by retro-orbital puncture on days 2, 4 and 6 post-infection for the determination of viral genome copies. Some animals (n = 5) from each group were euthanized on day 4 post-lethal challenge and organs (brain, liver and spleen) were harvested for determination of tissue viral load and immune responses. The remaining mice either died on days 6, 7 or 9 or were euthanized on day 10.

### 2.7. Tissue Harvest, Quantitation of Viral Genome Copy Numbers and Infectious Virus Titers

The brain, liver and spleen were harvested and virus genome copy numbers in tissue samples were determined using quantitative reverse transcription and polymerase chain reaction (qRT-PCR), and infectious virus titers were determined using plaque assay on monolayers of Vero cells, as detailed previously [53,54].

### 2.8. Preparation of Single Cell Suspension, Stimulation and Flow Cytometry

Spleens were harvested from mice and single cell suspension was prepared. Briefly, harvested spleens were minced and strained through 70 μm cell strainer. This was followed by washing with sterile cold PBS prior to resuspension of the cells in RBC lysis buffer (Biolegend catalog # 420301). The cells were washed again in sterile cold PBS and resuspended in RPMI media supplemented with 5% FBS and 1% antibiotics. Percentages of viable cells were determined by using trypan blue exclusion assay. Approximately 1 × 10^6^ cells were stimulated with 1 mg/mL of soluble DIII protein for 16 h prior to flow cytometric analysis.

Stimulated cells were stained using the following antibodies: fixable viability eFluor 780 (eBioscience catalog# 65-0865-14), anti-CD8α-PerCP/Cy5.5 (Biolegend catalog # 100733), CD4-Brilliant Violet 650 (catalog # 100545) and anti-IFNγ-eFluor 450 (Invitrogen catalog # 48-7319-42). Antibodies were diluted as per the manufacturer’s recommendation. Cells were resuspended in FACS buffer (PBS, 2% FBS) and flowcytometric analysis was performed on a BD Cytoflex flowcytometer. The results were analyzed using FlowJo software. In each analysis, respective FMO controls were used to set up the gates or to identify the positive populations. The gating strategy applied for the evaluation of flow cytometry-acquired data was as follows: Whole cells (SSC-A, FSC-A), Singlet population (FSC-H, FSC-A), Live cells (Viability-H, FSC-A), CD8α+ cells from live cell population (SSC-H, CD8α+-H) and IFNγ+ cells from CD8+ cells (CD8α+-H, IFNγ+ -H). Similarly, CD4+ cells from live cell population (SSC-H, CD4+-H) and IFNγ+ cells from CD4+ cells (CD4+-H, IFNγ+ -H) were generated.

### 2.9. PRNT_50_ Determination

The plaque reduction neutralization test (PRNT) was performed as described previously [57] to measure ZIKV neutralizing antibody titers. All serum samples were heat-inactivated at 56 °C for 30 min prior to testing. The serum samples were initially diluted 1:10 or 1:100 in PBS followed by further appropriate dilutions. A 50 µL volume of virus suspension containing ~200 PFU was mixed with an equal volume of various dilutions of serum samples and incubated at 37 °C for 1 h. The mixture was then used to infect monolayers of cells in 12-well plates at 37 °C for 1 h with rocking every 10 min. Following infection, the inoculum was removed, and the cells were overlaid with medium containing 1% low gelling temperature (LGT) agarose in VGM as described for the plaque assay. After incubation for 5 days at 37 °C, the plaques were counted manually. Antibody titers were determined as the reciprocal of the serum dilution that resulted in 50% reduction in the number of virus plaques (PRNT_50_), and geometric mean titers (GMT) were calculated from the PRNT_50_ values.

### 2.10. Antibody Isotyping

Antibody isotyping was performed as described previously [57]. Briefly, ELISA plates were coated with soluble ZIKV DIII protein at 2 µg/mL in PBS for 16 h at 4 °C. After addition in blocking buffer (Invitrogen Superblock, ThermoFisher Scientific, Waltham, MA, USA), serum samples diluted in blocking buffer (1:50) were added and incubated at room temperature for 2 h. After removal of serum samples, biotinylated goat antimouse IgG, biotinylated rat antimouse IgG1 or biotinylated rat antimouse IgG2a (Biolegend, San Diego, CA, USA; catalog #405303, #406603 and #407103, respectively) were added in blocking buffer at 1 µg/mL and incubated for 1 h. After three washes with PBS, streptavidin horseradish peroxidase was added (1:1000; BD-Pharmingen, San Diego, CA, USA). Following 30 min of incubation and three washes with PBS, 3,3-,5,5-tetramethylbenzidine-hydrogen peroxide solution was added to each well. IgG subclasses specific to anti-DIII protein antibodies were measured by determining optical densities at the 450 nm wavelength (OD450) with a reference OD570 using an ELISA plate reader.

### 2.11. Statistical Analysis

Data were analyzed using GraphPad Prism software version 6.0. Either the unpaired two-tailed Student’s t-test, Mann–Whitney test or Kruskal–Wallis test was performed for pairwise comparisons between the groups to determine significant differences in viral loads (RNA levels, infectious titer, etc.), clinical scores, PRNT_50_ and other immune response parameters. Pearson correlation between the relative levels of antibody and neutralization titers was examined using GraphPad Prism software. Data were represented as means ± standard error of mean (SEM). *p* values of ≤ 0.05 were considered significant.

## 3. Results

### 3.1. Construction of Ferritin Nanoparticle Vaccine Candidate Expressing Zika Virus Domain III (zDIII-F), Expression and Purification of the Protein/Nanoparticle

It has been reported that removal of the first four amino acids from the amino-terminus of ferritin and fusion of a protein immunogen at this terminus results in the exposure of the immunogen to the outside leading to better visibility of the antigen to BCRs [31]. In order to construct a ferritin nanoparticle displaying ZIKV envelope (E) immunogen, the coding sequence of ferritin (aa residues 5–167) from *Helicobacter pylori* along with other coding regions (described below) was synthesized using a commercial gene synthesis service (GenScript, Piscataway, NJ, USA) and was cloned into the CMV-promoter driven mammalian expression vector, pIRES2-eGFP, after removing the IRES-eGFP sequences. Since immunization with full length ZIKV E protein may contribute to antibody-dependent enhancement (ADE) of infection leading to more severe disease [48,49,50,51,52], only the domain III (DIII) of ZIKV E protein was used in our studies as it has been shown to elicit a strong neutralizing antibody response in mice without ADE [58]. Therefore, the coding sequence corresponding to the DIII spanning residues 303 to 404 of the ZIKV E protein from our infectious clone [53] was also fused immediately upstream of the amino-terminus of ferritin (Figure 1A). A flexible peptide linker (L) of four residues (Gly-Ser-Gly-Gly) was inserted between the DIII and ferritin sequences for proper folding of the two regions. To allow for efficient secretion of the fusion protein (zDIII-F) from transfected cells for ease of purification, we also inserted a consensus signal sequence, called secrecon (Sec) signal, at the amino-terminus of the fusion protein (Figure 1A).

Following sequence confirmation of the constructed plasmid, we examined expression and secretion of zDIII-F chimeric protein. HEK293T cells were transfected with the plasmid and culture supernatant and cell lysates were harvested and subjected to Western blotting to detect the zDIII-F protein. A protein band of approximately 27 kDa, consistent with the expected size of zDIII-F fusion protein, was detected in the supernatant (Figure 1B, lane 2) by using the ZV-57 monoclonal antibody directed against DIII of ZIKV E protein. This protein was also detected in the transfected cell lysates (lane 1). Interestingly, a slower migrating band of ~30 kDa was also observed in the cell pellet (lane 1), which is likely the signal sequence-uncleaved version of the fusion zDIII-F protein.

Since the zDIII-F protein could be readily detected in supernatants of transfected cells, we subsequently transfected a suspension culture of HEK293-F cells with the plasmid and purified zDIII-F from the culture supernatants collected at 4 days post-transfection using ion exchange chromatography followed by size exclusion chromatography. Although the ion-exchange chromatography resulted in enrichment of the zDIII-F protein (Figure 1C) in some pooled fractions, size exclusion chromatography yielded the protein that was greater than 90% pure, as judged by using Coomassie blue staining (Figure 1D). Western blotting using the ZV-57 antibody confirmed that the purified protein was indeed zDIII-F (Figure 1C,D). Transmission electron microscopic images of purified zDIII-F revealed the presence of spherical structures about 14–15 nm in diameter with projected structures indicating that the recombinant protein was able to form a nanoparticle-like structure with the DIII region likely protruding from its surface (Figure 1E). We observed that these nanoparticle structures are stable for at least ninety days when stored at 4 °C.

### 3.2. Enhanced Protection of Mice from ZIKV-Induced Diseases and Lethality Is Conferred by Vaccination with zDIII-F Nanoparticle

The vaccination and challenge strategy for testing the zDIII-F vaccine candidate is shown in Figure 2A. Three-week-old Ifnar1^−/−^ A129 mice were divided into three groups:

PBS group (n = 6), DIII group (n = 7) and zDIII-F group (n = 7). Mice were injected with either PBS (PBS group), 4 μg of soluble ZIKV DIII protein per animal (DIII group) or 10 μg of the zDIII-F nanoparticle vaccine (molar equivalent to 4 μg of DIII antigen) per animal (zDIII-F group) formulated with Addavax adjuvant. No adverse effects due to vaccination were observed in these animals. On day 28, mice in each group were challenged with 10,000 pfu of infectious clone-derived rMR766 ZIKV per mouse and monitored and scored for development of clinical signs of disease, weight loss and survival for 10 days, at which time the surviving mice were euthanized (Figure 2B). At 2–3 days post-challenge (dpc), the mice in the PBS group began showing signs of disease such as shivering and ruffled fur, which by day 3–4 progressed to conjunctivitis, lethargy and a hunched posture (Figure 2B). Additionally, some mice in this group also presented with both fore limb and hind limb paralysis beginning at 4 dpc which extended until day 6. All the mice in this group developed severe disease signs and succumbed to virus infection by 7 dpc with one mouse each dying on 5 dpc and 6 dpc (Figure 2B, PBS group). Mice vaccinated with soluble DIII (DIII group) appeared healthy for 3 dpc but three mice developed clinical symptoms exhibiting ruffled fur, conjunctivitis, followed by limb paralysis by day 6 and ultimately succumbing to infection by day 8. All other mice in this group were healthy until day 10, at which time they were euthanized (Figure 2B). In contrast, five out of seven mice in the zDIII-F group showed no symptoms of disease even at 10 dpc. Two mice presented mild symptoms with ruffled fur and shivering on 3 dpc but one mouse recovered completely while the other mouse exhibited more severe symptoms and succumbed on day 8 (Figure 2B). At the end of the experiment, on day 10 when they were euthanized, all mice (except the one that died on day 8) in this group appeared healthy.

Weight loss was observed in all mice administered with PBS and challenged with the rMR766 virus by 2 dpc (Figure 2C), and this trend continued until they succumbed to virus infection. While one mouse each from this group succumbed to infection on days 5 and 6, all other mice perished by 7 dpc (Figure 2D). In contrast, mice vaccinated with soluble DIII or zDIII-F and infected with the virus did not exhibit any weight loss. In fact, most mice in these groups began to gain weight after about 3 dpi and the trend continued up to 10 dpc (Figure 2C). Three mice from the soluble DIII-vaccinated group died on day 8, whereas six out of seven mice in the zDIII-F-vaccinated group survived the virus infection until 10 dpi when all the mice were euthanized (Figure 2D).

Viral genome copy numbers in the serum of the infected mice were significantly reduced in the soluble DIII-vaccinated and zDIII-F-vaccinated groups at 2 dpi in comparison to mice in the PBS group (Figure 2E). On days 4 and 6 post-challenge, significantly lower viral genome copy numbers were detected in the majority of the animals in the two vaccinated groups. On the other hand, high levels of viral genomes were detected in all animals administered with PBS. The same three mice from DIII-vaccinated and the same mouse from zDIII-F-vaccinated groups (that showed signs of clinical disease, Figure 2B) possessed high viral load (Figure 2E) at days 4 and 6, indicating that the deaths of these animals on day 8 may have been due to increased viral replication. These results show that administration of a single dose of the zDIII-F nanoparticle vaccine is sufficient to confer significant protection to the animals from lethal challenge. The protection conferred by DIII vaccination was less than that elicited by zDIII-F vaccination. Thus, the results suggest that the nanoparticle-based zDIII-F vaccine protects Ifnar1^−/−^A129 mice from lethal challenge and that zDIII-F appears to be a more efficacious vaccine candidate in protecting the animals and reducing viremia than the soluble DIII antigen.

### 3.3. Induction of Robust Neutralizing Antibody Response in zDIII-F-Vaccinated Animals

To examine if enhanced protection conferred by the zDIII-F nanoparticle is due to increased levels of neutralizing antibody response against ZIKV infection, we examined the presence of neutralizing antibodies in the serum samples of zDIII-F-vaccinated A129 mice collected at 28 days post-vaccination. Results show significantly higher neutralizing antibody titers (PRNT_50_ titer > 960) in sera from zDIII-F-nanoparticle-vaccinated animals in comparison to sera from animals from the PBS-administered group (Figure 3A). More importantly, the neutralizing antibody titers elicited by zDIII-F were significantly higher (>10-fold) than those elicited by the soluble DIII antigen (PRNT_50_ titer, ~90) (Figure 3A). These results indicate that vaccination with zDIII-F elicits a strong neutralizing antibody response that protects mice from lethal ZIKV challenge, whereas vaccination with soluble DIII confers protection to a lesser degree.

To determine if the antibodies induced by zDIII-F vaccination can neutralize different ZIKV strains from recent outbreaks, we performed a PRNT_50_ assay with the serum from zDIII-F-vaccinated mice. We used two contemporary isolates (PRVABC59 and MEX1–7) of ZIKV. These strains are of Asian lineage of ZIKV, which are different from the MR766 strain of East African lineage of the virus [59]. The results show potent inhibition of the ZIKV strains with similar PRNT_50_ values (Figure 3B), suggesting that the zDIII-F vaccine candidate may confer protection against other ZIKV strains.

Evaluation of IgG levels in the sera of animals from various vaccinated groups showed that, although there was a small but statistically significant increase in total IgG levels in the soluble zDIII-vaccinated group over that seen in PBS group, the ZDIII-F-vaccinated group showed significantly higher levels of total IgG compared to those seen in DIII or PBS group (Figure 3C). While no significant differences in the levels of IgG1 subclass of antibodies were seen between DIII and zDIII-F-vaccinated animals, IgG1 antibody levels were significantly higher than those from the PBS group (Figure 3D). Strikingly, the IgG2a subclass of antibodies were significantly higher in the zDIII-F group as compared to those from the DIII group or PBS group (Figure 3E). The increased levels of IgG2a in zDIII-F-vaccinated animals may indicate that a greater Th1-type immune response is conferred by the zDIII-F vaccine compared with the DIII vaccine and is likely involved in better protection of animals from virus challenge in this group. Pearson correlation analysis revealed that neutralizing antibody titers significantly correlated with the levels of IgG and IgG2a but not with that of IgG1 (Figure 3F).

### 3.4. Higher Frequency of IFNγ-Producing CD4 and CD8 T Cells Were Observed in Animals Vaccinated with zDIII-F as Compared to Those Vaccinated with DIII

Effector CD8 T cells and Th1 CD4 T cells are generated in response to ZIKV infection in the host [60] and play important roles in restricting ZIKV replication when type I IFN response is compromised [61]. Th1-mediated cellular immunity offers long-term immunological resistance against viral disease and works in concert with humoral immunity and memory B and T cells [62]. Several vaccines developed using ZIKV structural and non-structural proteins as antigens have been able to trigger unique T cell responses against the virus which assist in protective immunity and promote viral clearance [63,64,65]. Therefore, we next compared the T cell responses in animals vaccinated with DIII or zDIII-F and with those treated with PBS.

Groups (n = 6) of 6–7-week-old mice were vaccinated intramuscularly with either PBS, DIII or zDIII-F at doses described in Figure 2. The mice were sacrificed twenty-one days post-vaccination and the spleens were harvested. Splenocytes were stimulated with soluble DIII for 16 h, stained for viability, CD4, CD8 and intracellular IFN-γ, and then analyzed via flow cytometry. Results show that the mice vaccinated with the soluble DIII antigen exhibited a significant increase in the frequency of DIII-specific IFN*γ*-producing CD4 T cells as compared to those in the PBS control group (Figure 4A,B). More importantly, the mice vaccinated with zDIII-F showed a much stronger response and significantly higher frequency of IFN*γ*-producing CD4 T cells (mean percentage = 1.38%) compared to the PBS group (mean percentage = 0.23%) or the DIII group (mean percentage = 0.47%) (Figure 4A,B). Likewise, IFN*γ*-producing CD8 T cells were significantly higher in animals vaccinated with zDIII-F (mean percentage = 0.61%) as compared to those vaccinated with soluble DIII (mean percentage = 0.33%) or in the control group (mean percentage = 0.17%) (Figure 4C,D). Overall, these results suggest that both CD4 T cells as well as CD8 T cells expressing IFN-*g* are induced by both vaccine candidates, although the magnitude of response is much greater for CD4 T cells in mice vaccinated with zDIII-F.

### 3.5. Passive Transfer of Sera from zDIII-F-Vaccinated Mice Protects Animals from Lethal Virus Challenge

Since we observed a significant induction of neutralizing antibody response in animals vaccinated with zDIII-F nanoparticles, we explored the possibility that these antibodies, when transferred to naïve animals, could protect them from lethal ZIKV challenge. We conducted a passive antibody transfer experiment in A129 mice as shown in Figure 5A.

Mice were divided into two groups: one group (n = 10) received sera obtained from mice that were administered with PBS (naïve serum group), while the other (zDIII-F group) (n = 10) received sera from mice vaccinated with zDIII-F. These sera were collected from the animals at 28 days post-vaccination that had been injected with PBS or zDIII-F and the sera pooled from several animals were used in this experiment. Each mouse in the groups was administered 200 µL of pooled sera s.c. that had been heat-inactivated at 56 °C for 30 min. In similar passive transfer experiments [57], we had previously shown that within 2 h post-administration, sera collected from the animals following transfer contained neutralizing antibodies, indicating successful passive transfer of neutralizing antibodies by this route of inoculation. The mice were subsequently challenged with 10,000 pfu ZIKV at 4 h post-sera administration and were monitored for disease progression for 10 days (Figure 5B). The mice that received sera from PBS-administered animals developed disease, with at least 50% of them showing ruffled fur and shivering at 2 dpi. By 3 dpi, some mice developed conjunctivitis, lethargy and a hunched posture (naïve serum group, Figure 5B). Fifty percent of mice from this group were randomly sacrificed on day 4 to collect the brain, spleen and liver to determine viral load, while the others were left to monitor disease progression. In contrast, nine out of ten mice that received sera from zDIII-F-vaccinated animals appeared healthy until day 4 when 50% of mice (picked randomly) were sacrificed to collect the brain, spleen and liver to determine viral load, while the others were left to monitor disease progression. One animal from this group progressively exhibited symptoms of disease from day 3 and succumbed to infection on day 9 (Figure 5B). The remaining five mice that received sera from PBS-administered animals developed severe disease and by 6–7 dpi succumbed to the challenge (Figure 5B). Four out of the remaining five mice that received sera from zDIII-F-vaccinated mice did not exhibit any clinical signs of disease and survived (Figure 5B). It is not clear why one mouse from this group exhibited clinical signs of disease and succumbed to infection. It is possible that for reason(s) that are not understood at this time, this mouse may not have responded well to passive transfer of the antibodies. The mice that received sera from PBS-administered animals also began to lose weight, and by 4 dpc, weight loss was over 15%, while the mice in the other group continued to gain weight (Figure 5C). All animals in the naïve-serum-administered group perished by day 7, while 80% of animals (four out of five remaining mice) administered with zDIII-F serum survived with no clinical signs of disease until the day of termination of the experiment (Figure 5D).

Determination of viral genome copies in serum samples collected from mice in both groups showed that the mice receiving sera from zDIII-F-vaccinated animals had a significantly lower viral load in comparison to the mice that received the sera from PBS-administered mice (Figure 5E). It should be noted that the mouse from the zDIII-F serum-administered group that eventually succumbed to infection on day 9 had higher genome copy numbers on days 2, 4 and 6 (Figure 5E), indicating that the passive antibody transfer failed in this mouse, resulting in its death due to increased viral replication. Examination of viral genome copies in the brain, spleen and the liver of the animals sacrificed on day 4 revealed significantly reduced viral load in mice that received the sera from zDIII-F-vaccinated animals (Figure 5F) in comparison to those receiving sera from PBS-administered animals. These results demonstrate that passive transfer of sera obtained from mice vaccinated with zDIII-F confers solid protection against lethal ZIKV challenge, thus confirming that the development of neutralizing antibodies is a major correlate or indicator of protection against ZIKV.

## 4. Discussion

The emergence of ZIKV in the western hemisphere and its association with neurological disease in infants as well as adults poses a serious concern to public health. Licensed vaccines or antiviral drugs to treat ZIKV infections are not available, thereby necessitating the development of countermeasures against this viral epidemic. Although several different vaccine candidates have been tested [12,13,14], none have been approved for clinical use. Therefore, an efficacious vaccine is still urgently needed. As ZIKV infections have occurred in DENV endemic areas, and in view of the observations that cross-reactive antibodies can cause ADE [11,66,67], attention must also be paid to the development of safe and efficacious ZIKV vaccines that minimize or abrogate induction of ADE of DENV or ZIKV infections. Furthermore, due to the detection of ZIKV infections in many countries, particularly in resource-limited developing countries, rapid production of vaccines in a cost-effective manner is highly desirable.

We initially planned to use the ectodomain of E protein as the target for development of a nanoparticle vaccine candidate. Although neutralizing antibodies that are protective against ZIKV infections are mostly directed against the E protein, it has become clear that non-neutralizing antibodies generated against other epitopes in E protein have the potential to induce ADE of ZIKV and other flavivirus infections [68]. Studies have shown that antibodies induced by DI and DII regions of the E protein are poorly neutralizing and can mediate ADE [11]. Therefore, we focused on using the DIII domain of the ZIKV E protein, which has been shown to induce neutralizing antibodies that do not cause ADE [58,69,70,71].

Nanoparticles provide an improved vaccine platform for the delivery of antigens [16] and have been in clinical development against several infectious diseases [26,27,72]. Ferritin is a stable protein that has the ability to self-assemble into nanoparticles and this property has been exploited in the development of vaccine candidates against several pathogenic viruses. In the present study, we have developed a bacterial ferritin-based vaccine candidate using the ZIKV E DIII as the antigen. We have observed that the fusion protein (zDIII-F) self-assembles into a nanocage-like structure with the antigen likely displayed on its surface. The zDIII-F nanoparticles were found to be resistant to degradation and maintained structural integrity when left at 4 °C for up to 3 months, demonstrating a desirable property of thermostability of a vaccine candidate.

While designing our nanoparticle construct, we rationalized that adding a flexible linker between DIII and ferritin sequences would allow the DIII region to be displayed on the surface of the nanoparticle in its native structure and be recognized efficiently by B cells. However, attempts to determine its structure by cryo-EM were likely compromised due to the presence of the flexible linker sequences. Insertion of a less flexible linker sequence or direct fusion of DIII with ferritin may allow determination of the DIII structure that would be visible by the immune cells. Although our data show that this zDIII-F nanoparticle induced a robust immunological and protective response, it would be interesting to see if a construct such as the one without a flexible linker would induce a stronger response as compared to the one we have described here. The use of the consensus secrecon signal sequence in the plasmid construct for this vaccine candidate greatly enhanced secretion of the synthesized protein into the culture medium from which it could be purified in a simple and easily up-scalable purification process.

In the vaccination regimen used here, we have demonstrated that a single dose of zDIII-F nanoparticles induced significantly higher neutralizing antibody response as compared to the soluble DIII. Although we used only one dose (10 µg) of the nanoparticle vaccine, it is expected that varying the antigen dose and/or employing a prime-boost strategy may further enhance the immune responses. In a previous study, a ZIKV vaccine candidate displaying the E DIII domain on an immunologically optimized cucumber mosaic virus-derived VLP was shown to induce high levels of specific IgG after a single injection that could neutralize ZIKV in vitro [58]. It should be noted that most DIII-based vaccine candidates in the form of soluble antigen, nanoparticles, in tandem repeats or having been displayed on the surface of VLPs, were shown to induce neutralizing antibody responses to varying degrees that were protective against ZIKV [40,69,70,71,73,74]. However, an exception to this was reported in study [75], showing that DIII conferred limited protection on animals against the ZIKV challenge and suggested that the DIII may not be an appropriate vaccine candidate for induction of protective immune responses against ZIKV. Although the reason(s) for this discrepancy is not known at this time, it should be noted that the DIII is considered weakly immunogenic and the authors of the study used more conventional vaccine approaches such as DNA, protein and adenoviral-vectored vaccines [75]. The nanoparticle vaccine approach in which multiple copies of an antigen are displayed on the surface of the particles offers substantial improvement in eliciting robust neutralizing antibody responses even when a weakly immunogenic protein is used. Our results presented here, demonstrating that zDIII-F induces high levels of protective neutralizing antibody responses, are consistent with studies using nanoparticle approaches.

In a recent study, Rong et al. [40] described a nanovaccine using the DIII region of the E protein of an Asian lineage ZIKV. The nanovaccine was based on a self-assembling human heavy chain ferritin molecule that presented the DIII on its surface and was shown to confer complete protection against lethal challenge with the homologous virus. The authors further demonstrated a lack of ADE of infection of DENV. Although the nanovaccine was found to induce robust humoral (PRNT_50_ titer of >160) as well as cellular immune responses following two booster administrations without adjuvant [40], it appears that the vaccine candidate described in our studies here is more potent in inducing stronger humoral responses (PRNT_50_ titer of ~960), even with a single administration. Whether the difference is due to the vaccination regimen, the platform, the expression system, the use of adjuvant or other factors is not known and requires further investigation.

Studies have shown that ferritin nanoparticles induce robust protective humoral and cellular immune responses against a number of viral pathogens, including influenza virus, EBV, HIV-1, SARS-CoV-2 and others [31,37,38,39,40,41,42,43,44,45,46,47]. Consistent with these findings, we have shown that immunization of mice with ferritin-based nanoparticles (zDIII-F) displaying ZIKV E DIII elicits strong antigen specific neutralizing antibody responses, as well as cell-mediated responses, which correlate with the protection of animals against lethal ZIKV challenge. We have observed significantly higher levels of total IgG and IgG2a with zDIII-F, which may indicate that elevated Th1-type cellular response likely contributes to the higher levels of IgG2a specific antibody class. This interpretation is consistent with the observed increase in CD4+/IFN-γ+ T cells in zDIII-F-vaccinated animals compared to DIII-vaccinated animals. Indeed, it has been shown that induction of ZIKV-specific CD4+/IFN-γ+ T cells is also critical to B cell activation, antibody maturation and protection against ZIKV infection [76]. In contrast, CD8+/IFN-γ+ T cells could protect the animals by eliminating the ZIKV-infected cells as has been demonstrated previously [77]. Our observation that both CD4+/IFN-γ+ T cells and CD8+/IFN-γ+ T cells were induced significantly by the zDIII-F as compared to the soluble DIII antigen, suggests that the robust protection conferred by vaccination with the zDIII-F nanoparticle is due to both antibody-mediated as well as cell-mediated immune responses.

The neutralizing potency of zDIII-F-induced antibodies against two ZIKV isolates (PRVABC59 and MEX1-7, both belonging to the Asian lineage) from recent outbreaks was found to be very similar to that against the homologous strain (rMR, belonging to East African lineage), whose DIII sequence was used for vaccine construction. The results suggest that the zDIII-F vaccine candidate will likely confer protection against challenge with African as well with Asian strains of the virus.

Overall, our studies presented here suggest that both humoral and cellular responses are likely contributing factors for protection induced by the zDIII-F vaccine candidate. We realize the limitations of our studies, such as the use of a single dose of the zDIII-F vaccine candidate, single immunization regimen and examining the efficacy in immunocompromised mice. Therefore, one must be cautious in extrapolating the data from the mice model to ZIKV infections in humans. Nevertheless, this study strengthens the possibility of developing an efficacious ZIKV vaccine that uses the E protein DIII as the antigen in a ferritin nanoparticle providing a multivalent delivery platform for enhanced immunological response against the virus.

## 5. Conclusions

Results presented in this study show that a ferritin nanoparticle-based vaccine candidate with domain III of ZIKV E protein as the immunogen induces significantly higher levels of both humoral and cell-mediated immune responses in mice as compared to the soluble form of DIII. Mice immunized with this nanoparticle vaccine candidate are robustly protected from lethal challenge with the virus. The high levels of immune responses and protection conferred by this vaccine candidate provide opportunity for further studies and clinical evaluation as a potential vaccine against ZIKV.

## Figures and Tables

**Figure 1 vaccines-11-00821-f001:**
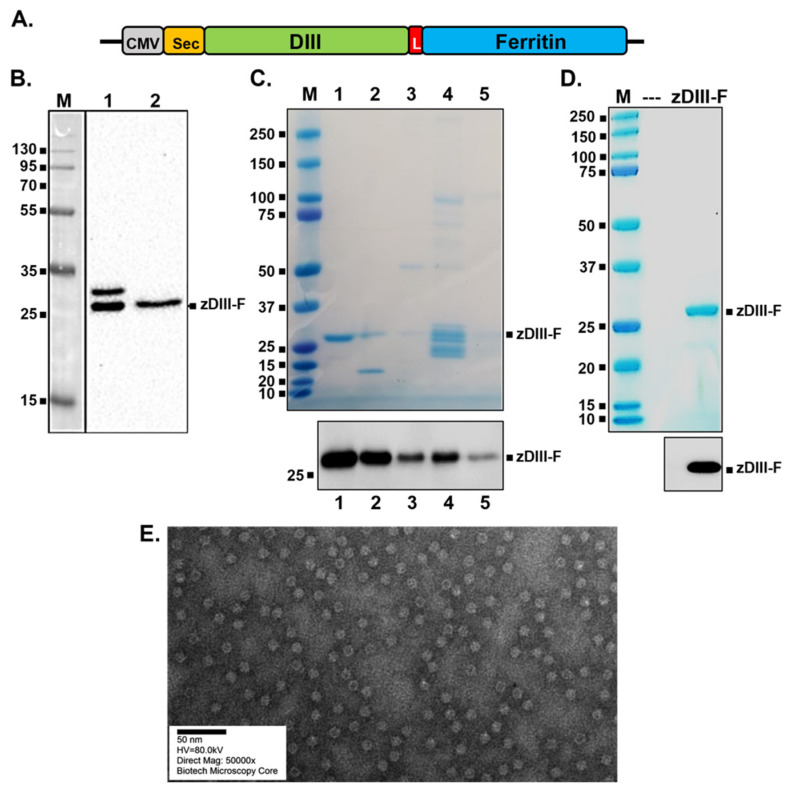
Construction, expression and purification of zDIII-F protein and nanoparticles. (**A**) The schematic of the plasmid-encoding zDIII-F. The relevant elements of the plasmid are shown in boxes with the vector backbone represented as a thick black line. CMV, cytomegalovirus promoter in pIRES2-eGFP vector; Sec, secrecon signal sequence; DIII, MR766 ZIKV domain III spanning residues 303–404; L, linker sequences; Ferritin, ferritin sequences spanning residues 5–167. (**B**) Expression of zDIII-F in HEK293T cells. Cells were transfected with the above plasmid and 96 h after transfection, cell lysates (lane 1) and supernatants (lane 2) were analyzed by SDS-PAGE followed by Western blotting using the monoclonal antibody ZV-57. Molecular masses (in kDa) of marker proteins are shown on the left. The secreted zDIII-F protein of ~27 kDa is identified on the right. The slower migrating protein in lane 1 likely corresponds to the zDIII-F protein with its signal sequence still attached. (**C**) SDS-PAGE and Coomassie staining (top) and Western blot analysis with ZV-57 antibody (bottom) of various pooled fractions from anion exchange column chromatography. Eluted zDIII-F protein was detected using Coomassie staining (top) in pooled fractions of the gradient with concentrations of NaCl ranging as follows: 107–158 mM (lane 1), 159–214 mM (lane 2), 215–271 mM (lane 3), 272–379 mM (lane 4) and 380–422 mM (lane 5). Approximately 100 ng of total protein from each of the pooled fractions was analyzed for Western blot analysis (bottom). Molecular masses of marker proteins are shown on the left side of each panel. (**D**) SDS-PAGE followed by Coomassie staining (top) and Western blot analysis (bottom) of zDIII-F after purification using size exclusion column chromatography. Approximately 1 µg of total protein for Coomassie staining (top) and 100 ng of total protein for Western blotting (bottom) were analyzed. (**E**) TEM image of purified zDIII-F nanoparticles. Scale bar, 50 nm.

**Figure 2 vaccines-11-00821-f002:**
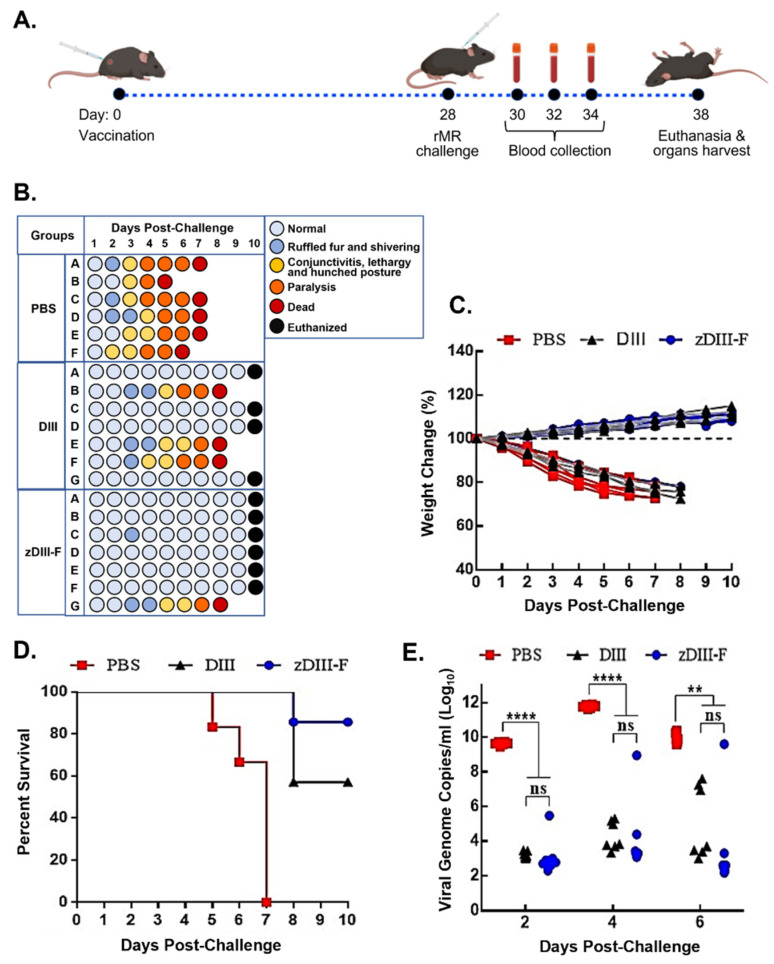
Vaccination with zDIII-F protects mice from lethal ZIKV challenge. (**A**) Scheme of vaccination and challenge experiment. Groups of animals were vaccinated on day 0 and challenged with rMR ZIKV on day 28. Blood samples were collected on days 30, 32 and 34 (or 2, 4 and 6 days post-challenge). All surviving mice were euthanized at the end of the experiment on day 10. (**B**) Clinical symptoms in individual animals (identified as A to F/G in the groups) at various days following ZIKV challenge. (**C**) Percentage change in weight of individual animals in various groups following challenge with ZIKV. (**D**) Percent survival of animals in various groups following ZIKV challenge. (**E**) Viral genome copies in serum of animals on days 2, 4, and 6 post-challenge as measured by qRT-PCR. Data presented as mean ± SEM. ns, non-significant; **, *p* ≤ 0.01; and ****, *p* ≤ 0.0001.

**Figure 3 vaccines-11-00821-f003:**
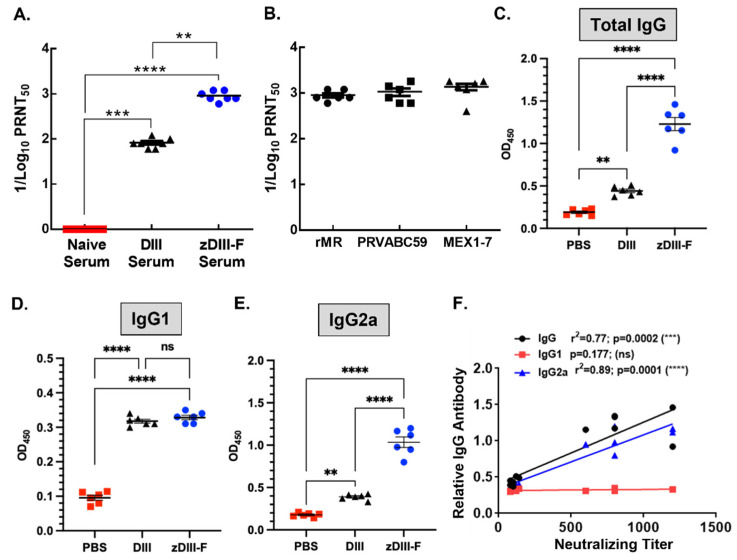
Neutralizing antibody titers in the sera of vaccinated animals. (**A**) ZIKV neutralizing antibody titers in the sera expressed as reciprocal of PRNT_50_ from naïve, soluble DIII- or zDIII-F-vaccinated mice at 28 days post-vaccination. (**B**) ZIKV neutralizing antibody titers (PRNT_50_) in the sera of zDIII-F-vaccinated animals against two Asian lineage (PRVABC59 and MEX1-7) ZIKV isolates. Quantitation of total IgG (**C**), IgG1 (**D**) and IgG2a (**E**) in the sera of individual animals in various groups at 28 days post-vaccination. (**F**) Pearson correlation of antibody levels and neutralizing titers. ns, non-significant; **, *p* ≤ 0.01; ***, *p* ≤ 0.001; and ****, *p* ≤ 0.0001.

**Figure 4 vaccines-11-00821-f004:**
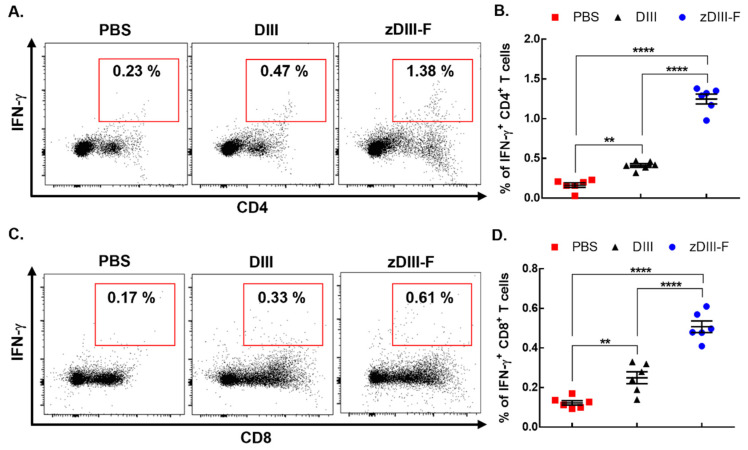
zDIII-F vaccine candidate induces potent cellular immune response in vaccinated mice. BALB/c mice (n = 6) were administered either PBS, DIII or zDIII-F. Spleens were harvested at 21 days post-vaccination and spleenocytes were stimulated with DIII and stained for CD4, CD8 and IFN-γ. Representative flow cytometry dot plots are shown for CD4+IFN-γ+ cells (**A**) and CD8+IFN-γ+ cells (**C**). Data from (**A**,**C**) are presented in plots shown in panels (**B**,**D**), respectively. Unpaired Student’s t-test (two-tailed) was used to determine significance between the groups. Data are presented as mean ± SEM. **, *p* < 0.01; ****, *p* < 0.0001.

**Figure 5 vaccines-11-00821-f005:**
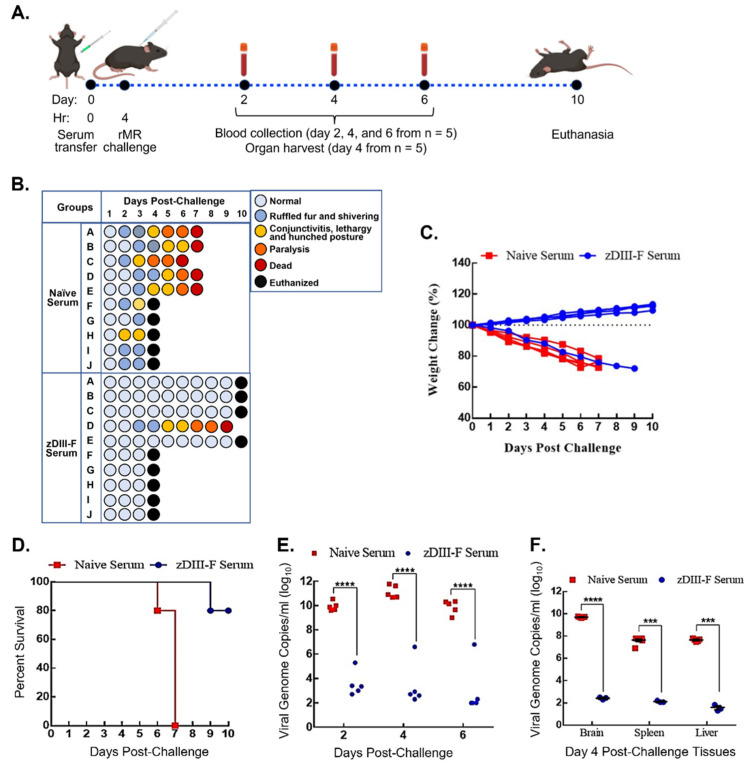
Passive transfer of sera from zDIII-F-vaccinated animals protects naïve animals from lethal ZIKV challenge. (**A**) Scheme for passive transfer of immune sera and lethal ZIKV challenge in four-week-old A129 mice. Mice were injected on day 0 with 200 µL of PBS (n = 10) or pooled sera from zDIII-F-vaccinated animals (n = 10) s.c. at 0 hr and challenged with rMR (10,000 pfu/mouse) at 4 hr. Blood samples were collected on days 2, 4 and 6 post-lethal challenge from five animals in each group. Organs were harvested from the other five animals on day 4. All surviving animals were euthanized on day 10. (**B**) Clinical symptoms in individual mice (identified as A to J in each group) following lethal ZIKV challenge are shown. Percent weight change (**C**) and survival (**D**) of mice injected with immune sear from naïve or zDIII-F-vaccinated animals. (**E**) Viral genome copies in sera of animals (n = 5) in each group collected on days 2, 4 and 6 post-challenge. (**F**) Viral genome copies per gram of tissue (brain, spleen and liver) on day 4 post-challenge (n = 5) as measured by qRT-PCR. Data are presented as mean ± SEM. Two-way ANOVA was used to determine significance in panels (**E**,**F**). ***, *p* ≤ 0.001; ****, *p* ≤ 0.0001.

## Data Availability

The data presented in this study are available on request from the corresponding author.

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
