# Peer review of "A Ferritin Nanoparticle-Based Zika Virus Vaccine Candidate Induces Robust Humoral and Cellular Immune Responses and Protects Mice from Lethal Virus Challenge"

_vaccines, 2023, doi:10.3390/vaccines11040821_

Round 1

Reviewer 1 Report

The paper by Pattnaik described a vaccine candidate based on the EDIII domain of Zika virus displayed by ferritin nanoparticles. While the author show nice neutralisation titters and protection from infection in mice there are 2 fundamental problems with the paper:

1) It is inconsistent with the neutralisation data that the ELISA titers are so low; indeed they seem barely measurable. There seems to be a fundamental problem.

2) The nanoparticle really is barely better than the EDIII domain alone. This may be due to the adjuvant used but it appears the authors used the same adjuvant with the particles. So there really is little benefit for the nanoparticle. The experiments should be done without adjuvant.

Author Response

We appreciate the reviewer for the helpful comments on the manuscript. The following are our responses to the reviewer’s concerns/comments:

  1. “It is inconsistent with the neutralization data ...” We agree with the reviewer that ELISA titers (OD450) are low, but it should be noted that the original serum samples were diluted 1:50 and the values are significantly higher in the DIII, or zDIII-F vaccinated animals as compared to the PBS control group. We repeated the ELISA with the frozen sera from these animals using a fresh antibody kit from the manufacturer and the results are included in the revised manuscript. Although the ELISA titers have improved, the trend remained essentially similar to those obtained previously. The antibody titers in animals in DIII and zDIII-F vaccinated groups are significantly higher than the PBS group.
  2. “The nanoparticle really is barely better ...” We respectfully disagree with the reviewer’s contention here since the PRNT50 titer for the nanoparticle vaccine candidate (zDIII-F) is significantly higher (~ 10 fold) than that with DIII vaccine candidate. Additionally, cell-mediated immune responses were also significantly higher with zDIII-F as compared to DIII alone. We understand the limitations of our study, some of which are discussed in the manuscript (line 691, pp. 17). In future studies, we will address some of those limitations including conducting the experiment without adjuvant as suggested by the reviewer.

Reviewer 2 Report

The authors in this article show promising results in a nanoparticle vaccine fused to DIII domain of E protein with ferritin.  The article shows Relevant results for the field of research. Some minor revisions are needed to improve the text.

In the abstract, the author uses "appended" to define the nanoparticle, is better to use direct "fused".

At the beginning of the introduction "and" is used three times in the same line, try to use another word.

In the section on material e methods- plasmid constructs, please include the restriction sites used in each step.

Why the amount of 10 micrograms was directly used? Did the authors try different concentrations before? Is important to show that large amounts of antigen do not produce ADE effect.

In Fig.3(B), why only the serum from the nanoparticle was used? The serum from DIII immunization should be used in the same experiment.

Author Response

We thank the reviewer for appreciating promising results of our nanoparticle vaccine studies and its relevance to the field of research. The following are our responses to the minor comments made by the reviewer:

  1. “In the abstract, the author uses ...” Changed as suggested.
  2. “At the beginning of the introduction “and” is used three times ...” Changed as suggested.
  3. “In the section on materials and methods ...” Restriction sites have been included.
  4. “Why the amount of 10 micrograms ...” We chose this dose based on reports in the literature using similar vaccine candidates. We did not try different doses and acknowledge this as one of the limitations of our study (described in the original manuscript, line 691, pp. 17). Future studies may address some of these limitations keeping in mind the possibility of ADE although DIII as a vaccine candidate has not been shown to induce ADE.
  5. “In Fig. 3(B), why only the serum from ...” Since the zDIII-F vaccine candidate induced significantly higher neutralizing activity, we only tested the sera for neutralization activity against strains from recent outbreaks. Future studies may include DIII sera for testing the neutralization activity.    

Reviewer 3 Report

The manuscript is well written and with good data set presentation. Several questions still needed to be addressed before publication.

1. Figure 1. E shows the intact ferritin nanoparticle structure. I am wondering whether you have any symmetry studies for the ferritin nanoparticle structure.

2. Figure 2. D still shows less than 20 percent mortality.  The dose in the challenge with ZIKV is rMR, 10,000 pfu/mouse, so what the corresponding LD50 is here? Maybe less LD50 dose can keep a 100% survival rate?

3. Figure 3. C-E. What level of serum dilution is used here? If the OD450 is less than 0.2, then it can be considered as having no neutralizing activity. Thus the DIII group sera merely had no antiviral activity since the total IgG OD450 is less than 0.2.  Do you have the serial dilution data for different groups?

4. Figure 2. C could be adjusted by changing the legend color or shape since it is very hard to discern from different groups.

Author Response

We thank the reviewer for commenting that the “manuscript is well written and with good data ...” The following are our responses to the reviewer’s comments:

  1. “Figure 1. E shows the intact ferritin nanoparticle ...” Since our initial attempt to determine the structure of the zDIII-F nanoparticles was unsuccessful, we did not conduct additional studies.
  2. “Figure 2. D still shows less than 20 percent mortality ...” We used a lethal dose based on our previous studies so that the efficacy of the vaccine candidate could be readily ascertained. We did not determine the LD50 in our studies. We concur with the reviewer that reducing the dose of the virus in challenge studies may lead to 100% survival, but the unvaccinated control animals may also survive the challenge, compromising the data interpretation.
  3. “Figure 3. C-E. What level of serum dilution ...” 1:50 dilution of sera was used for antibody quantitation. We understand the reviewer’s concerns regarding the OD450 Therefore, we repeated the ELISA with the frozen serum samples from this experiments with the same dilution using a fresh kit from the manufacturer and have found that the OD450 values are improved. The new data showing that the antibody titers in animals in DIII and zDIII-F vaccinated groups are significantly higher than the PBS group have been included in the revised manuscript. Importantly, we conducted Pearson’s correlation analysis of the relative levels of antibody titers and neutralizing activity. The results (included in the revised manuscript, Fig. 3.F) show that levels of IgG and IgG2a correlate (r2= 0.77 and .89 for IgG and IgG2a, respectively) significantly (p ≤ 0.0002) with neutralizing activity whereas IgG1 levels do not.
  4. “Figure 2. C could be adjusted by changing the legend color or shape ...” Different colors and shapes were used for animals in different groups. Since individual animal weight changes were plotted, the graph appears hard to discern. However, the trend for the animals in each group can be clearly seen. We did not change the colors or symbols in this graph since the individual anima data may still overlap.   

Round 2

Reviewer 1 Report

The paper is fine now